# Neural Network-Enabled Identification of Weak Inspiratory Efforts during Pressure Support Ventilation Using Ventilator Waveforms

**DOI:** 10.3390/jpm13020347

**Published:** 2023-02-16

**Authors:** Stella Soundoulounaki, Emmanouil Sylligardos, Evangelia Akoumianaki, Markos Sigalas, Eumorfia Kondili, Dimitrios Georgopoulos, Panos Trahanias, Katerina Vaporidi

**Affiliations:** 1Department of Intensive Care Medicine, School of Medicine, University of Crete, 71003 Heraklion, Greece; 2Institute of Computer Science, Foundation for Research and Technology—Hellas (FORTH), 70013 Heraklion, Greece; 3Department of Computer Science, University of Crete, 70013 Heraklion, Greece

**Keywords:** machine learning, assisted ventilation, ventilator-induced diaphragmatic disfunction, flow waveform, monitoring

## Abstract

During pressure support ventilation (PSV), excessive assist results in weak inspiratory efforts and promotes diaphragm atrophy and delayed weaning. The aim of this study was to develop a classifier using a neural network to identify weak inspiratory efforts during PSV, based on the ventilator waveforms. Recordings of flow, airway, esophageal and gastric pressures from critically ill patients were used to create an annotated dataset, using data from 37 patients at 2–5 different levels of support, computing the inspiratory time and effort for every breath. The complete dataset was randomly split, and data from 22 patients (45,650 breaths) were used to develop the model. Using a One-Dimensional Convolutional Neural Network, a predictive model was developed to characterize the inspiratory effort of each breath as weak or not, using a threshold of 50 cmH_2_O*s/min. The following results were produced by implementing the model on data from 15 different patients (31,343 breaths). The model predicted weak inspiratory efforts with a sensitivity of 88%, specificity of 72%, positive predictive value of 40%, and negative predictive value of 96%. These results provide a ‘proof-of-concept’ for the ability of such a neural-network based predictive model to facilitate the implementation of personalized assisted ventilation.

## 1. Introduction

Diaphragm disuse atrophy is a well-recognized complication of mechanical ventilation contributing to ventilator-induced diaphragmatic dysfunction (VIDD) [1,2,3]. Diaphragm disuse atrophy was initially described in patients during passive ventilation [4], but it has since been shown that VIDD may also develop during assisted ventilation, when assist is excessive and patient’s effort very weak [3,5]. The development of VIDD has important clinical implications as it has been shown to affect patients’ outcomes [4,6]. 

Avoiding weak inspiratory efforts from excessive assist during assisted ventilation, essential to avoid VIDD, necessitates the monitoring of inspiratory efforts. Such monitoring in critically ill patients can be performed using esophageal pressure, but it requires dedicated equipment and can be technically challenging and time-consuming, so it is not routinely implemented [7]. In current clinical practice there is no other non-invasive method for the continuous monitoring of inspiratory effort. The respiratory frequency (f), tidal volume (VT) or the breathing pattern (f/VT) may only indicate excessively high or low effort at extremes values, such as a respiratory rate below 12 or above 30 breaths/min, while the P0.1 can intermittently provide an indication of low respiratory drive but has many limitations, and requires manual inspection [8,9]. 

During pressure support ventilation (PSV) the shape of inspiratory flow depends on the pressures generated by the patient’s respiratory muscles (Pmus) and the ventilator (Pvent) and the respiratory system mechanics. The change in the curvature of inspiratory flow to an exponential decay, similar to that of passive, pressure-regulated ventilation, can be visually identified and can indicate the time point of relaxation of the patient’s inspiratory muscles during mechanical inspiration [10]. Weak efforts are often short in duration, so the shape of flow waveform may be used to identify weak efforts and potentially excessive ventilatory assist. Clinicians are currently advised to inspect the flow waveform during PSV and try to decrease assist if a passive-like flow shape is present. An automated algorithm within the ventilator recognizing a risk of excessive assist could facilitate the titration of assist during PSV, improving efficiency, decreasing the caretakers’ workload, and likely contributing to better patient outcomes.

In this study we employ a Deep Few-Shot Learning technique, namely a One-Dimensional Convolutional Neural Network (1D CNN), to automatically detect key features of the flow waveform during PSV to identify weak inspiratory efforts in critically ill patients. The network was trained on annotated breaths in which effort was quantified by the pressure-time product of the respiratory muscles (PTP), using a threshold of 50 cmH_2_O*s/min to characterize weak efforts. 

## 2. Materials and Methods

This study was performed using recordings obtained for clinical purposes during the past 5 years (2018–2022) at the ICU of the University Hospital of Heraklion. We used a dataset consisting of recordings during PSV which included measurements of flow, airway, esophageal and gastric pressures, with a duration of at least 5 min. Several of these recordings have been included in the analysis of previously published studies [11,12]. The clinical indications for these recordings were mainly titration of assist and/or evaluation of inspiratory effort and the diaphragm’s contribution to tidal breathing. The use of the anonymized data has been approved by the hospital’s ethics committee. The clinical characteristics, demographics and ventilation variables were obtained from the medical records. Respiratory system compliance was computed manually from end-inspiratory occlusions obtained at the higher level of assist assuring by the inspection of esophageal and gastric pressures waveforms passive conditions.

### 2.1. Development of the Annotated Dataset

To train and test the network we developed an annotated dataset of all recordings, in which the effort of each breath was quantified and invalid breaths were removed. The first step included the calculation of Pmus and the construction of the Pmus over time waveform, followed by an algorithm for the automated identification of inspiratory time and the calculation of PTP (Figure 1A). To identify the mechanical inspiratory time (Ti) and segment the mechanical inspirations from the dataset a wavelet-based technique was employed, where the start of inspiration was set where the rapid decline of Paw began at the beginning of the breath (triggering) and the end of inspiration was identified as the point where flow was zero at the transition from positive to negative (Figure 1B). The pressure-time product (PTP) of the Pmus was calculated as the integral of the Pmus curve over time during inspiration (PTP per breath) and multiplied by the respiratory frequency to obtain PTP per minute. Automated calculation of the PTP was performed for each mechanical inspiration identified using the wavelet-based technique, and compared with the PTP manually computed by three experts using the AcqKnowledge software [13]. 

To ensure the system’s accuracy, we validated the annotated dataset at three levels.

Segmentation: Correct identification of the inspiratory time was visually assessed by an expert (K.V.) using dedicated software (Figure 1). The automated detection of mechanical inspiration was characterized as correct or incorrect, and expressed as percent (correct over total). Periods of time during the recording where accurate measurement of esophageal pressure could not be obtained (for example, during esophageal contractions or cough) were identified and excluded from further analysis.

PTP Computation Assessment: the calculated PTP values per breath and per minute were quantitatively assessed against manually computed PTP values. Specifically, a random sample of breaths (8 breaths per level of assist) was selected for manual annotation (PTP computation) by three experts (S.S., E.A. and K.V.), and the corresponding 8-breath median was compared against that of the estimated one. 

Classification Assessment: The PTP per min was classified as low or not, using as threshold a value of 50 cmH_2_O×s/min, as suggested by the current literature [5,9,16]. The classification of PTP/min as low or not based on the computed PTP/min was assessed against the manually computed PTP classification, and expressed as the percentage of correctly classified breaths having or not a low PTP/min.

### 2.2. Development of the 1D-CNN to Identify Weak Efforts

The architecture of the employed 1D-CNN shown in Figure 2A has been previously described in detail [17]; however, the model was restructured and retrained using the Pmus. More specifically, the model is composed of (i) a 2D input layer, fed with Flow and Paw segments, (ii) five 1D convolutional blocks with two identical layers each, to extract representative patterns used for classification, (iii) two dense layers, to increase model’s complexity and enhance its generalization capacity, and (iv) the output layer. The number of filters and the kernels’ size, respectively, for the convolutional layers are (64, 3), (64, 9), (128, 9), (256, 9), (32, 7), while the two dense layers consist of 128 neurons and have a dropout of 0.5. The classification outcome is provided by the output layer, which consists of a single neuron and sigmoid activation. The model is designed to receive a single segmented breath and provide a single number (output) between 0 and 1 that characterizes the probability of the effort in this breath to be weak (1 being highest probability). The model requires no information regarding the settings of the MV or the physiological state of the patient. To develop the model the dataset was randomly split into training, validation and test sets using data from different patients in each set. The annotated data of the training set were used to train the model, while the data of the validation set were used to finetune the hyper-parameters of the model. Finally, the test set was used to provide the final results. In the test set, the classification was based only on Flow and Paw waveforms, while the computed PTP was used to categorize results as true or false positives or negatives. The block diagram is shown in Figure 2B.

### 2.3. Statistical Analysis 

Patients’ demographics, clinical and ventilation characteristics are presented as percentiles, means and standard deviation (SD), or medians and interquartile range (IQR), depending on the distribution of values. The validation results are presented as a percentage of correct classifications, and as absolute differences between the two computations. The 1D-CNN model classification of the PTP as low or not was compared with the computed PTP and the results were categorized as true or false positives or negatives. Sensitivity, specificity, positive (PPV) and negative predictive value (NPV) and accuracy were computed using standard formulas. 

## 3. Results

### 3.1. Patients’ Characteristics

We analyzed the recordings of 37 patients, a total of 123 different levels of assist, corresponding to an average of three levels in each patient (range of two to five levels of assist per patient, 0–20 cmH_2_O of pressure support). The characteristics of the patients included in the analysis are presented in Table 1. At the time of the recording, RASS was −3 to −5 in 20 patients and −2 to 0 in 17. The median total duration of mechanical ventilation was 13.5 days (IQR 8–23 days), ICU stay was 16 days (IQR 10–25 days), and ICU mortality was 16%. All patients were ventilated using a Drager Evita XL ventilator, and the triggering delay range was 80–100 msec. The cycling-off criterion was at the default setting of 25% of peak flow. Low levels of intrinsic PEEP (<1 cmH_2_O) were present in seven patients, and premature cycling off was observed in one patient. The PTP values observed in the complete dataset had an IQR of 77–197 cmH2O×s/min (median 136 cmH_2_O*s/min, Figure 3), and 14% were classified as weak, having a PTP/min < 50 cmH2O*s/min. 

### 3.2. Validations of the Annotated Dataset

The validation of inspiration segmentation, performed for all breaths, showed that in 99.9% of the breaths the algorithm correctly identified the beginning and end of the mechanical breath. The validation of PTP computation showed that the median difference between the manual and computed PTP/min was 4 cmH2O×s/min (IQR = 1–9 cmH_2_O×s/min). The computed PTP/min was correctly classified as low or not in all but one case (99.2% correct), taking the manually computed PTP/min classification as reference.

### 3.3. Predictive Model Results

A total of 76,993 breaths was included in the analysis. The training and validation sets included 45,650 breaths from 22 patients; the tests set included 15 patients and 31,343 breaths. The results presented correspond to the analysis of the 15 patients. The model predicted a weak inspiratory effort with a sensitivity of 88%, specificity of 72%, PPV 40%, NPV 96%, and accuracy of 75% (Figure 4). Among breaths incorrectly classified as weak, 59% had a PTP less than 100 cmH2O×s/min (Figure 5). 

## 4. Discussion

The neural network-based classifier developed and validated in this study identifies breaths with weak inspiratory efforts non-invasively, using the flow waveform during PSV. Combined with the wavelet-based method proposed for the segmentation of the respiratory signals, it provides an end-to-end approach which could be implemented as a bedside ‘smart alarm’, using the signals of flow and pressure obtained from the ventilators. The model has high sensitivity, so it can accurately exclude weak inspiratory efforts, which is important since clinical examination lacks such sensitivity and cannot discriminate weak from normal efforts. If this classifier were to be implemented as an alarm for weak inspiratory efforts during PSV, as observed 88% of weak efforts would be correctly identified. While 28% of normal-effort breaths would be characterized by the model as weak, in more than half of those the actual level of effort would be in the low-to-normal range (with a PTP/min equal or less than 100 cmH_2_O×s/min), suggesting that a trial of lower assist would be appropriate. 

The identification of weak inspiratory efforts during PSV is important for critically ill patients for two reasons. Firstly, because PSV is one of the most commonly used modes of assisted ventilation, and secondly because while excessive assist and weak efforts can occur often, they can hardly be detected. In the case of a high level of support and relatively normal mechanics, even weak inspiratory efforts may result in adequate VT, thus placing the patient at risk of VIDD. To avoid excessive assist during PSV, pressure support should be adjusted to the minimum required. However, this titration is not so easily accomplished in everyday practice, because there are no clinical signs discriminating normal from weak inspiratory efforts, and thus adequate from excessive assist [18,19]. This is a major limitation of PSV, likely contributing to the increased risk of asynchronies and delayed weaning observed in this mode [20,21,22,23]. Indeed, the respiratory rate, proposed as the non-invasive parameter with the greatest accuracy for diagnosing over-assist, was shown in one study to have a sensitivity of 80% when using a threshold of 17 br/min while a threshold of 12 br/min had a sensitivity of 100% [18]. In our dataset a threshold of 17 br/min had a sensitivity of 70% in identifying weak efforts, while a threshold of 12 br/min had a sensitivity of 88% (as did our predictive model), although such a low respiratory rate was observed in only 5% of all breaths. 

Inspiratory effort can be continuously monitored by measuring the esophageal pressure; however, there are unaddressed technical challenges in this method, such as the correct placement and filling of the catheter, so the use of esophageal pressure monitoring during assisted ventilation is not widely used in everyday practice. Other non-invasive indices such as P0.1 and occlusion pressure can intermittently provide an estimation of inspiratory drive and effort, and have been shown to predict excessive effort. Apart from efforts to optimize and automate the measurement of esophageal pressure, other non-invasive methods to continuously monitor patient effort are needed. The first and obvious ‘candidate’ non-invasive variable to be used as an index of inspiratory effort during PSV is the inspiratory flow.

The shape of inspiratory flow during PSV depends on Pmus, and thus provides information on the duration of respiratory muscle contraction during the mechanical inspiration. It is textbook knowledge that a passive-like inspiratory flow is often associated with weak inspiratory effort and excessive assist, and should be managed by decreasing the level of pressure support [24]. Relying on visual inspection of the ventilator screen as a way of continuously monitoring patients’ inspiratory effort is clearly impractical, and thus the need for other automated methods and alarms arises.

A recent study evaluated the use of flow waveform during PSV to estimate patients’ effort and introduced the flow index [19,25], a number describing the concavity of the inspiratory flow waveform. This study included data from 24 patients, at three different levels of assist each, and 702 breaths in total. The ventilatory characteristics of the patients, including the levels of pressure support, range of effort and respiratory system compliance, were similar to our study. The flow index was shown to have a sensitivity of 73%, specificity of 70%, PPV of 76%, and NPV of 66% to detect weak inspiratory efforts. The validation of the flow index in a dataset from brain injury patients showed higher sensitivity, but the range of inspiratory effort was lower and respiratory system mechanics better in that dataset [26]. This is important, as the identification of weak effort based on the flow shape becomes less accurate as mechanics worsen. The classifier developed in our study performed better than the flow index (having a sensitivity of 88 vs. 73% of the flow index) in a patient population in whom titration of assist was challenging for the clinicians. 

The computational methodology, as well as the clinical characteristics of the population the data were collected from, are important for the evaluation of the final model. The selected Deep Learning schema for the detection of key features of the flow waveform is most suitable for the aim of the study, since the convolutional kernels are specifically designed to identify such types of patterns that separate two classes. The shallow architecture and the use of weighted loss mitigate against the relatively small and unbalanced dataset, respectively, both of which are common issues that occur when applying deep learning algorithms to medical data [27]. Additionally, the computation method used to calculate the PTP, based on which the model was trained, was compared to experts’ manual analysis and found to be accurate. The final model developed was validated using a dataset of recordings from patients not used in the development phase. The dataset used in the study included breaths having a relatively broad range of inspiratory efforts (as shown in Figure 3), obtained using the range of pressure support levels used in everyday practice. Other factors that may affect the model’s performance are the conditions affecting the correlation between patient effort and inspiratory flow waveform, which include the triggering delay, the respiratory system mechanics, and the presence of asynchronies such as ineffective efforts and premature cycling off. The triggering delay and cycling-off criterion were those typically encountered in most critically ill patients. The range of respiratory system compliance was 46 ± 14 mL/cmH_2_O. Although resistance could not be measured, none of the patients in the study had severe obstructive lung disease and low levels of PEEPi were found only in seven out of the thirty-seven patients. Ineffective efforts, not included in the computation of PTP/min, were observed in two patients at the high assist level. A premature cycling-off asynchrony was observed in one patient, in whom the manually computed PTP differed from the one calculated using the automated algorithm. Overall conditions such as PEEPi, ineffective efforts and premature cycling off, which affect the correlation between patient effort and the shape of inspiratory flow, were rarely present in this dataset. 

Machine learning (ML) approaches are increasingly being implemented to address problems related to mechanical ventilation. The rationale in all such studies is the need for automation and smart alarms, as continuous monitoring of the ventilator screen by expert clinicians is not realistic. Most of these studies have focused on the identification of asynchronies during mechanical ventilation [28], with excellent results. Several ML-based methods have been used [29,30], including more recently a two-layer Long Short-Term Memory (LSTM) network and a 1D-CNN [31,32]. The benefits of neural network-based approaches rely on the inherent ability of a NN to recognize patterns and extract features from raw data without extensive pre-processing and hardcoded feature engineering, which makes them particularly appealing for implementation in monitoring devices. Convolutional deep neural networks have been used to extract features from ECG to identify arrhythmias [33], predict atrial fibrillation [34] and hyperkalemia [35] with promising results. Algorithms to detect asynchronies or weak inspiratory efforts, such as the one developed in this study, could be introduced in ventilators’ software or in monitoring devices connected to the ventilators.

Some limitations of our study are that it is a single center study, using one type of ventilator and data acquisition system, and a relatively small sample size. The quality of the ventilator waveform data and the triggering delay are now similar in most commercially available ICU ventilators and would not be expected to affect the reproducibility of the model. We addressed the problem of a relatively small sample size by designing a classifier that can generalize to never-seen patients, thus ensuring that our methodology is robust and extensible. Nevertheless, further validation and optimization of the model in larger datasets, which would include patients with a broader range of respiratory system mechanics, shape and magnitude of respiratory muscle pressure, is necessary. Patients were not prospectively recruited just for the purpose of the study, but rather were patients who presented some challenges in titration of assist, which prompted clinicians to measure esophageal and gastric pressures. Thus, they represent the patient population in whom such an algorithm would be useful. Furthermore, we tried to adequately describe the clinical and ventilatory characteristics of the patients, to enhance the clarity of the data source. It is important to note that patients with severe obstructive lung disease, in whom a large proportion of their breathing effort may be consumed just to trigger the ventilator, were not included in this study. In such patients, models evaluating the effort remaining after triggering would likely misclassify efforts as weak. Finally, our study and others in the current research field are limited by the lack of a threshold of weak inspiratory effort that is proven to be clinically significant.

## 5. Conclusions

In conclusion, the results of this study show the feasibility of a 1D-CNN-based classifier to evaluate the ventilator flow waveform during PSV, and classify inspiratory efforts as weak or not with high sensitivity and high negative predictive value. The results of this study, together with similar studies on ML-based algorithms for the detection of asynchronies, pave the way for the implementation of ML-based smart alarms to optimize patient effort and patient-ventilator interactions during assisted ventilation.

## Figures and Tables

**Figure 1 jpm-13-00347-f001:**
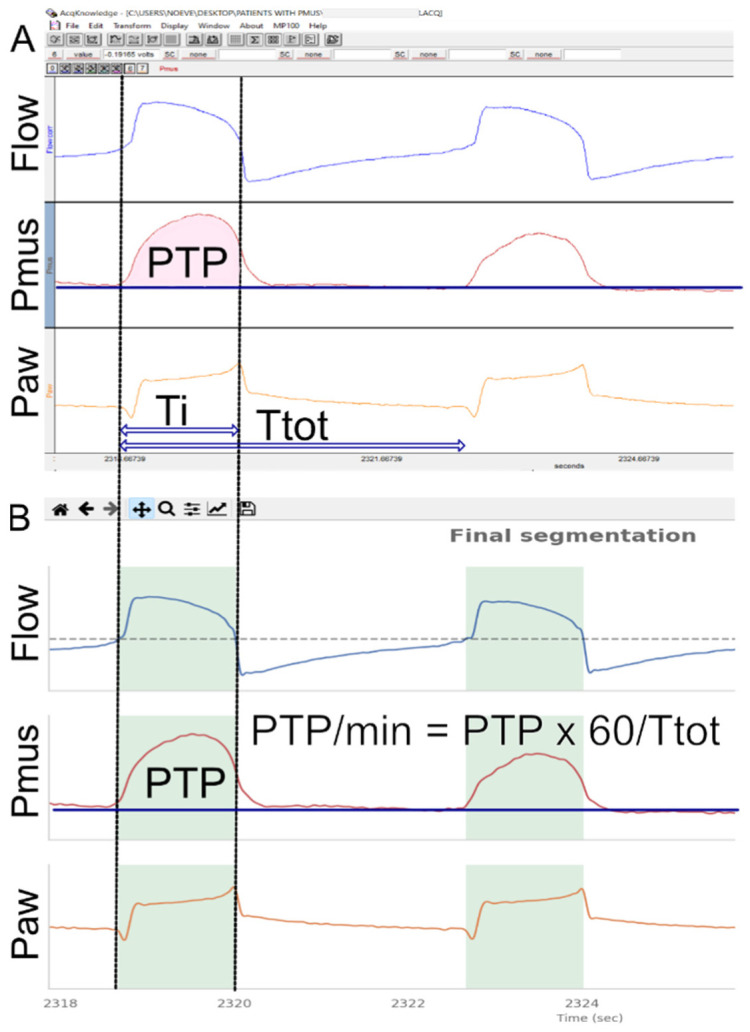
Identification of inspiratory time and computation of the respiratory muscle pressure pressure-time product. Flow, respiratory muscle pressure (Pmus) and airway pressure (Paw) waveforms visualized using both the AcqKnowledge (**A**) and the developed software (**B**). The inspiratory time as identified using a wavelet-based technique is indicated by the green shaded area in panel B. Pmus was calculated as Pmus = (VT × Ecw) + (Rcw × Flow) + Pes_endexp_ − Pes, where tidal volume (VT) was computed as an integral of flow waveform, chest wall elastance (Ecw) was calculated from the predicted value of vital lung capacity (VC) [14,15] (Ecw = 0.025 × VC where VC was calculated from the equation: VC = (27.63 − 0.112 × age) × height(cm), for males and VC = (21.78 − 0.101 × age) × height(cm) for females), chest wall resistance, Rcw = 1.5 cmH_2_O/L, and the end-expiratory esophageal pressure (Pes_endexp_) was obtained from the recordings after confirming there was no contraction of expiratory muscles. The pressure-time product (PTP) of the Pmus was calculated as the integral of the Pmus curve over time during inspiration (PTP per breath), indicated here by the pink shaded area in panel A and multiplied by the respiratory rate to obtain PTP per minute. Ti: inspiratory time (mechanical), Ttot: total breath time.

**Figure 2 jpm-13-00347-f002:**
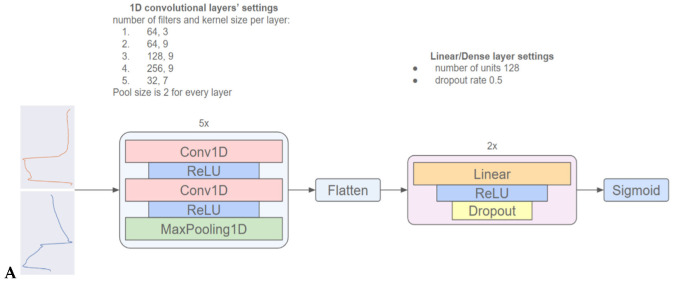
(**A**)—Model Architecture: The One-Dimensional Convolutional Neural Network (1D-CNN) consists of 5 convolutional blocks and 2 densely-connected layers. Each convolutional block applies a 1D convolution and a Rectified Linear Unit (ReLU) activation function twice, and then a 1D max pooling operation. The output of the convolutional blocks is flattened and sent to the densely-connected layers, which are composed of a linear transformation, a ReLU activation function and a dropout operation. Finally, the probability distribution for the two output classes is generated by a single neuron with a sigmoid activation function. (**B**)—Block diagram.

**Figure 3 jpm-13-00347-f003:**
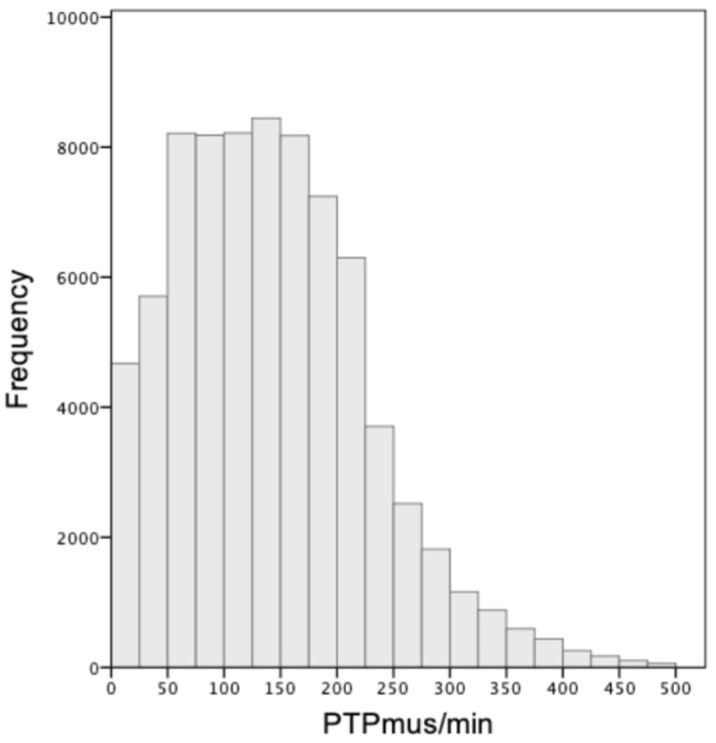
Range of inspiratory efforts. Distribution of the observed values of the respiratory muscle pressure pressure-time product per min (PTP/min) of the complete dataset, expressed as absolute numbers.

**Figure 4 jpm-13-00347-f004:**
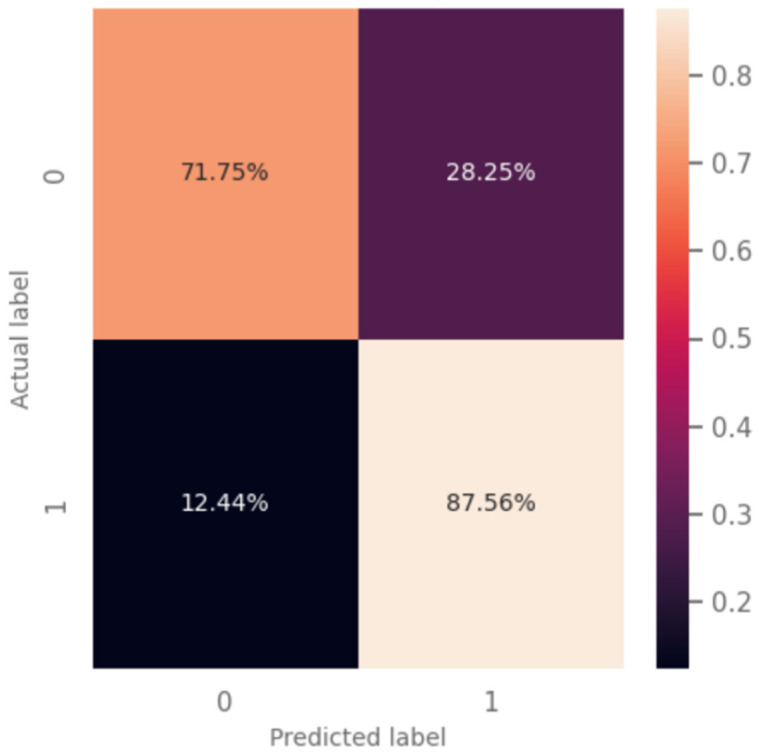
Model results. Confusion matrix of the model’s evaluation results using the data from 15 ‘never-seen’ patients. Weak efforts are labeled as ‘1’, while non-weak efforts are labeled as ‘0’. The optimal threshold of 0.42 was selected using the validation set.

**Figure 5 jpm-13-00347-f005:**
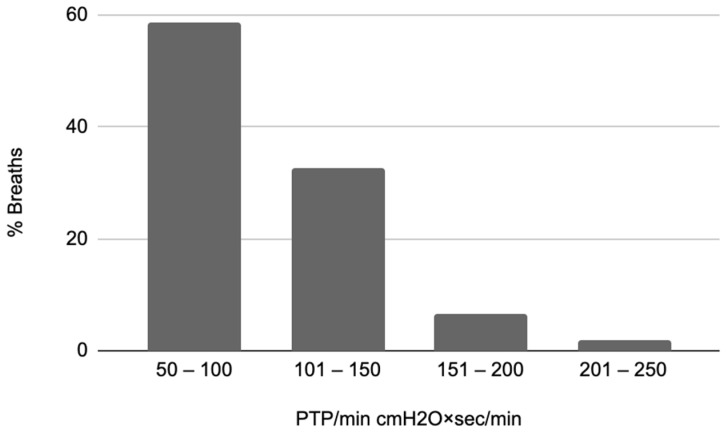
Distribution of pressure pressure-time product (PTP/min) values of breaths incorrectly classified as weak (false positives, FP), expressed as % of total FP.

**Table 1 jpm-13-00347-t001:** Patients’ characteristics.

Baseline characteristics	Age (years)	70 ± 10
	Males	46%
	BMI	30 ± 8
	APACHE-II score	19.3 ± 7.5
Admission diagnosis	Hypoxemic RF	38%
	Hypercapnic RF	19%
	Septic Shock	19%
	Acute brain injury	24%
Ventilation characteristics	Duration of MV prior to recording (days)	9 (IQR 5–12)
	PEEP cmH_2_O	7 ± 2
	FiO_2_	35% (IQR 30–40%)
	PaO_2_/FiO_2_	252 ± 63
	VT mL/kg IBW	7.7 (IQR 6.9–9.2)
	RR br/min	21 ± 5
	VE L/min	9.6 ± 1.6
	Pressure support cmH_2_O	7 ± 2
	Respiratory system compliance * mL/cmH_2_O	46 ± 14

Patients’ baseline characteristics and ventilation variables just prior to the recordings. Data are presented as percentages, means ± standard deviation or medians with interquartile range (IQR). Abbreviations: BMI: body mass index; APACHE-II: Acute Physiology and Chronic Health Evaluation-II; RF: respiratory failure; MV: mechanical ventilation; FiO_2_: fraction of inspired oxygen; PaO_2_: partial pressure of arterial oxygen; VT: tidal volume, IBW: ideal body weight; RR respiratory rate per minute; VE minute ventilation. * In three patients the respiratory system compliance could not be computed.

## Data Availability

All data included in this study can be provided by the corresponding author upon reasonable request.

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
