# Peer review of "Neural Network-Enabled Identification of Weak Inspiratory Efforts during Pressure Support Ventilation Using Ventilator Waveforms"

_jpm, 2023, doi:10.3390/jpm13020347_

Round 1

Reviewer 1 Report

-          Line 42- Authors said that there is no other non-invasive method can accurately estimate inspiratory effort, although in the discussion they cited the Flow index. Furthermore, Bertoni et al published an important paper in this field that should be cited (https://doi.org/10.1186/s13054-019-2617-0)

-          Paragraph between lines 59-65 is not necessary. Introduction can be shortened.

-          The model used Flow waveform to predict a weak inspiratory effort, but this information is clear, only in the discussion section. This should be clearly described in method section. Furthermore, what measure of flow waveform was used? Area? Shape?

-          How this model can be used? The algorithm will be installed in the ventilator?

Author Response

Thank you for your valuable comments and suggestions. After careful consideration of the reviewers' recommendations, we have performed the requested changes which we believe have significantly improved our manuscript. Please find below a point by point response to your comments, also marked red in the submitted revised manuscript. 

  1. Line 42- Authors said that there is no other non-invasive method can accurately estimate inspiratory effort, although in the discussion they cited the Flow index. Furthermore, Bertoni et al published an important paper in this field that should be cited (https://doi.org/10.1186/s13054-019-2617-0)

Response: thank you for this suggestion, we have corrected this sentence as “XXX” and included the following statement in discussion “XXX” along with the suggested citation, about the use of Pocc to predict excessive effort.

  1. Paragraph between lines 59-65 is not necessary. Introduction can be shortened.

Response: we have shortened introduction, trying to also incorporate Reviewer’s 2 suggestions.

  1. The model used Flow waveform to predict a weak inspiratory effort, but this information is clear, only in the discussion section. This should be clearly described in method section. Furthermore, what measure of flow waveform was used? Area? Shape?

Response: In the last paragraph of introduction it is stated: “In this study we employ a … to identify weak inspiratory efforts in critically ill patients.” Moreover, according to reviewer’s suggestion, we have modified in methods the sub-header as “Development of the 1D-CNN to identify weak efforts”

  1. How this model can be used? The algorithm will be installed in the ventilator?

Response: Thank you for this comment, indeed the algorithm could be incorporated in the ventilator’s software, or in a separate monitoring device connected to the ventilator. We have now included this in discussion: “Algorithms to detect asynchronies or weak inspiratory efforts, such as the one developed in this study, could be introduced in ventilators’ software or in monitoring devices connected to the ventilators.”

Reviewer 2 Report

#Write the motivation, novelty and contribution clear in the Introduction section. Write section organization at the end of the Introduction section. #Write detailed related work section. Highlight making and breaking of previous study. Highlight the research gap. #Justify the selection of Deep Few-Shot Learning technique. Read and cite: challenges of deep learning in medical image analysis-improving explainability and trust, Medical Big Data and internet of medical things: Advances, challenges and applications etc . #Add an object process diagram in the method section.  Write detailed discussion on method. #Result section is very poor. Compare with existing work. #Write limitations in conclusion section.

Author Response

Thank you for your valuable comments and suggestions. After careful consideration of the reviewers' recommendations, we have performed the requested changes which we believe have significantly improved our manuscript. Please find below a point by point response to your comments, also marked red in the submitted revised manuscript. 

  1. Write the motivation, novelty and contribution clear in the Introduction section.

Response: in the revised introduction section we have emphasized the motivation, novelty and contribution of the study, while trying to shorten introduction as per Reviewer’s (1) suggestion.

Motivation and contribution: “Avoiding weak inspiratory efforts from excessive assist during assisted ventilation, essential to avoid VIDD, necessitates monitoring of inspiratory effort.” and “An automated algorithm within the ventilator recognizing a risk of excessive assist could facilitate titration of assist during PSV, improving efficiency, decreasing  the caretakers’ workload, and likely contributing to better patient outcomes.”

Novelty: “In this study we employ a Deep Few-Shot Learning technique, namely an 1-Dimensional Convolutional Neural Network (1D CNN), to automatically detect key features of the flow waveform during PSV to identify weak inspiratory efforts in critically ill patients.”

  1. Write section organization at the end of the Introduction section.

Response: To align with reviewer 1 suggestion to shorten introduction, and because section organization is not typically included in “clinical” papers, we kindly suggest to maintain the current format of introduction.

  1. Add an object process diagram in the method section.

Response: we have included a block diagram as suggested.

  1. Write detailed related work section. Highlight making and breaking of previous study. Highlight the research gap. Justify the selection of Deep Few-Shot Learning technique. Read and cite: challenges of deep learning in medical image analysis-improving explainability and trust, Medical Big Data and internet of medical things: Advances, challenges and applications etc . Write detailed discussion on method. Write limitations in conclusion section. Result section is very poor. Compare with existing work.

Response: We have expanded the discussion, within the word limits suggested by the journal, and included the suggested ref. We believe the revised discussion better addresses the related work, and research gap, and the method used in the study. A native (American-English) speaker revised the manuscript (minor language editing is not marked red) . 

Round 2

Reviewer 1 Report

All my converns were corrected. 

Reviewer 2 Report

Resolved my queries.